# OpenReview forum: "System Prompt Extraction Attacks and Defenses in Large Language Models"
_ICLR.cc/2026/Conference — ICLR 2026 Conference Withdrawn Submission_

### Official Review · Reviewer_E1LY · 2025-10-29

**Soundness:** 2
**Presentation:** 2
**Contribution:** 3
**Rating:** 4
**Confidence:** 3

**Summary:**

This paper tackles the safety concern of system prompt extraction attacks in LLMs. The authors propose SPE-LLM, a framework for evaluating such attacks and corresponding defenses. The framework uses datasets of synthetic or publicly collected system prompts as controlled ground truth for evaluation. Novel adversarial queries are designed to extract system prompts from SOTA models accurately, and multiple defense strategies are proposed, with evaluation conducted across benchmark datasets using diverse metrics.

**Strengths:**

1. The paper addresses an important safety concern, where attackers may extract and exploit LLM system prompts.


2. The paper proposes improved adversarial query techniques (CoT, few-shot, extended sandwich) that extract system prompts more precisely than previous methods.

**Weaknesses:**

1. The formatting of the paper could be improved for better readability. The texts in Figure 3, Table 2 and 3, etc. are too small to read.

2. The writing of the introduction lacks clarity. It might be improved by including a concrete example on why system prompts extraction is a critical safety issue, in addition to citing the possible consequences.

3. It is not clear how the synthetic prompts represent/resemble real, even proprietary system prompts from both open- and closed-source LLMs. While these prompts follow previous works, justifications on this point may improve the significance of this work.

4. In addition, synthetic datasets may not reflect the diversity or complexity of real system prompts in commercial LLMs, limiting external validity.

**Questions:**

1. The paper mentions that the proposed defenses are limited. Could the authors elaborate on how adversarial instruction fine-tuning could strengthen defenses against system prompt extraction attacks?

2. How would fine-tuning attacks themselves impact system prompt security, and do they pose additional risks or limitations to the proposed defense strategies?

---

### Official Review · Reviewer_Z27w · 2025-11-01

**Soundness:** 3
**Presentation:** 3
**Contribution:** 2
**Rating:** 4
**Confidence:** 2

**Summary:**

This paper studies the system prompt extraction attacks and defenses in LLMs. System prompts often contain sensitive information and are vulnerable to extraction attacks. This paper considers a black-box treat model and use prompt engineering methods (CoT, Few-shot-prompting, and extended sandwich attack) to perform SPE attacks and defense. Experiments on multiple models are presented to validate the main claims.

**Strengths:**

1. (Originality) This paper propose a novel framework of SPE, which is different from most jailbreaking attacks.
2. (Clarity) The presentation of this paper is straightforward. The methods are easy to understand.

**Weaknesses:**

1. The significance of SPE attacks is not widely understood (compared to other jailbreaking attacks), nor is it discussed in detail in this paper. While the authors provide a list of references (cf. Lines 41-44) to support their arguments, the inclusion of specific, detailed examples would improve the illustrative power of this paper.
2. There is a lack of evaluation regarding the helpfulness and efficiency of the models after the defense mechanism has been integrated.

**Questions:**

1. How to evaluate the SPE attacks against closed sourced models? I cannot understand the rationale behind Lines 342-344.
2. Does the defense mechanism proposed in this paper exhibit any negative side effects? It is commonly believed that there is a tradeoff between the helpfulness and the robustness of LLMs.

---

### Official Review · Reviewer_Y3nK · 2025-11-01

**Soundness:** 2
**Presentation:** 2
**Contribution:** 2
**Rating:** 6
**Confidence:** 4

**Summary:**

This paper explores how a general-purpose assistant (Claude) can serve as a practical environment for alignment research. It proposes a new benchmark suite (Must-Do, Mustn’t-Do, and Should-Do) grounded in real assistant behaviors, and introduces evaluation protocols using sparse, high-quality human annotations. The authors aim to bridge academic alignment objectives with real-world assistant behavior and advocate for broad collaborative efforts in alignment research.

**Strengths:**

Here are the strengths of the paper:

Originality: Introduces a practical, behavior-grounded framework for alignment that reflects actual deployment use cases—distinct from abstract, synthetic benchmarks.

Clarity and Scope: Clearly articulates alignment categories (Must-Do, Mustn’t-Do, Should-Do) with illustrative examples and user-focused evaluation criteria.

Significance: Offers a valuable bridge between alignment theory and production-scale systems, encouraging open research collaboration grounded in realistic settings.

**Weaknesses:**

Here are the weakness of the paper:

Lack of rigorous experimentation: The paper emphasizes framework design and philosophical positioning, but offers limited empirical results or baselines for benchmarking.

Sparse evaluation detail: The use of “sparse but high-quality” annotations is advocated, but without detailed methodology for ensuring inter-rater reliability or statistical robustness.

**Questions:**

How do you ensure the consistency and reliability of the sparse human annotations used in your evaluations, particularly across nuanced alignment categories like “Should-Do”?

---

### Official Review · Reviewer_5a2F · 2025-11-05

**Soundness:** 2
**Presentation:** 2
**Contribution:** 1
**Rating:** 2
**Confidence:** 3

**Summary:**

The paper presents a framework called SPE-LLM that investigates system prompt extraction attacks and corresponding defenses. Specifically, it (1) designs new adversarial query strategies for extracting hidden system prompts, (2) proposes several defense techniques including instruction defense, sandwich defense, and system-prompt filtering, and (3) introduces evaluation metrics and conducts extensive experiments across multiple datasets and LLMs (Llama-3, Falcon-3, Gemma-2, GPT-4, GPT-4.1). The results show that the proposed defenses, particularly system-prompt filtering, can significantly reduce attack success rates.

**Strengths:**

This paper addresses the problem of system prompt extraction in large language models and investigates several defenses against such attacks. The topic is practically important, as system prompts often contain sensitive or proprietary instructions, and understanding their vulnerability is relevant for LLM deployment security.

**Weaknesses:**

While the topic of system-prompt extraction has high practical importance, the paper’s attacks and defenses largely replicate prior findings from 2023–2024 jailbreak and prompt-leakage literature. The experiments are thorough but incremental, offering quantitative confirmation rather than conceptual novelty.

**Questions:**

How does SPE-LLM differ fundamentally from prior prompt-injection or jailbreak frameworks beyond scale and dataset variety?

---

### Note · Authors · 2025-11-18

I have read and agree with the venue's withdrawal policy on behalf of myself and my co-authors.